# Dietary Plant Protein Intake Can Reduce Maternal Insulin Resistance during Pregnancy

**DOI:** 10.3390/nu14235039

**Published:** 2022-11-26

**Authors:** Yuting Hong, Chen Yang, Jinjing Zhong, Yanmei Hou, Kui Xie, Linlin Wang

**Affiliations:** 1Institute of Reproductive and Child Health/National Health Commission Key Laboratory of Reproductive Health, Department of Epidemiology and Biostatistics, School of Public Health, Peking University, Beijing 100191, China; 2Ausnutria Hyproca Nutrition Co., Ltd., Changsha 410000, China

**Keywords:** dietary protein, plant protein, pregnancy, insulin resistance, metabolomics

## Abstract

Evidence suggests that the source of dietary protein may have an impact on insulin resistance, but no studies have explored it in pregnant populations. In this study, we combined a population study and an animal experiment to explore this effect. The population study was conducted with data from NHANES. Multiple linear regression was used to observe the association of protein intake with outcomes, including fasting glucose (GLU), insulin (INS), and HOMA-IR. In the animal experiment, 36 pregnant SD rats in three groups were orally administered 100% animal protein, 50% animal protein and 50% plant protein, or 100% plant protein, respectively. The intervention continued throughout the whole pregnancy. On day 19.5, maternal plasma was collected after overnight fasting, and metabolomics was performed using UPLC-MS. We found plant protein intake was negatively correlated with INS and HOMA-IR in the whole population. During the third trimester, a similar correlation was also observed. The animal experiment also presented the same result. In metabolomic analysis, changes in various metabolites and related pathways including FoxO and mTOR signaling pathways were observed. In conclusion, we found a negative association between dietary plant protein intake and maternal insulin resistance during pregnancy. Changes in some active substances and related metabolic pathways may play an important role.

## 1. Introduction

Insulin is one of the core hormones regulating energy and substance metabolism. The decrease in insulin sensitivity or the increase in insulin resistance is considered to be related to various metabolic diseases. Current research suggests that the occurrence of insulin resistance is an important risk factor or early link for many diseases such as type 2 diabetes, hyperlipidemia, hypertension, and coronary heart disease [1,2,3]. During pregnancy, the mother will develop physiological insulin resistance, which will gradually increase with the development of pregnancy [4,5,6].

Studies have found increased insulin resistance in a variety of pregnancy disorders, including gestational diabetes mellitus and preeclampsia [7,8]. Insulin resistance during pregnancy is also thought to be associated with poor placental function and intrauterine growth restriction and is also thought to predict the occurrence of adverse pregnancy outcomes [7,9,10]. Maintaining a normal state of insulin resistance during pregnancy may help reduce the occurrence of these diseases and adverse outcomes, thereby benefiting the health of the mother and the fetus. Multiple genetic and environmental factors are thought to influence insulin sensitivity [11], and diet and exercise are considered to be important controllable factors that influence insulin sensitivity [3,12,13,14].

Different food compositions in the diet have been found to be associated with diabetes and insulin resistance in many studies. Red meat intake in a diet has been found to increase the risk of diabetes and insulin resistance in studies based on populations from various regions [15]. The effects of milk or dairy products on insulin resistance or diabetes are still contradictory [16]. In contrast, many dietary patterns rich in plant-derived foods have been found to benefit the control of diabetes and insulin resistance [17,18]. Several studies explored the relationship between dietary food compositions and gestational diabetes mellitus [19,20] and indicated that the intake of animal-derived foods could increase the risk of developing gestational diabetes mellitus. At the same time, specific nutrients in the diet were also believed to have a potential effect on insulin resistance. As dietary fat has long been found to affect insulin resistance [21], more recent research showed the importance of protein in influencing insulin resistance [22,23,24,25,26].

Protein is an important nutrient for pregnancy, and maternal protein requirements increase during pregnancy and are higher in late pregnancy [27,28]. However, studies in non-pregnant people have found that total dietary protein intake was positively associated with insulin resistance levels [22,23], which indicated that the regulation of insulin might be impacted by dietary protein. With the increased protein requirements and the development of physiological insulin resistance during pregnancy, it is even more important to study the effect of protein sources on insulin resistance during pregnancy to avoid adverse health outcomes for the mother and her offspring.

In recent years, many studies have gradually revealed the role of amino acids as signaling molecules [29]. For example, branched-chain amino acids (BCAAs), a class of amino acids thought to be more abundant in proteins of animal origin, are believed to potentially affect a variety of diabetes-related metabolic pathways [30,31,32]. In addition, certain amino acids rich in proteins of plant origin, such as arginine, were thought to have a beneficial effect on the body’s insulin metabolism [33,34]. The amino acid composition is considered to be very different between plant and animal proteins [35,36], which may provide a mechanistic basis for their influence on insulin resistance. Meanwhile, though such studies have revealed that the source of protein in the diet may also affect the regulation of insulin in the non-pregnant population [24,25,26], these relationships in the pregnant population are unclear due to the limited amount of data, while the mechanism is still not well understood.

This study aims to explore the effects of protein intake from different sources on maternal insulin secretion and insulin resistance by combining population and animal studies. Further, it aims to explore the possible mechanisms of these effects by applying high-throughput metabolomics methods.

## 2. Materials and Methods

The overall design of this study is shown in Figure 1.

### 2.1. Study Population

A population study was conducted with data collected from the National Health and Nutrition Examination Surveys (NHANES, https://www.cdc.gov/nchs/nhanes, accessed on 31 August 2021). Details on the implementation of the NHANES survey can be found in the relevant manuals (https://www.cdc.gov/nchs/nhanes/about_nhanes.htm, accessed on 31 August 2021).

We obtained demographic characteristics, dietary, examination, laboratory, and questionnaire data on all pregnant females from NHANES 1999–2018. Due to data availability, only pregnant females aged 20–44 who completed at least one dietary survey in MEC were included in the study.

Reproductive health and demographic characteristics were focused on in this study. Reproductive health characteristics include gestation period and parity. The gestation period was categorized as the first trimester (months 1 to 3), the second trimester (months 4 to 6), and the third trimester (months ≥ 7). Parity was defined as primiparous and non-primiparous. Demographic characteristics include age, race, education, and family income. A detailed classification of demographic characteristics can also be found in the corresponding data description of NHANES (https://wwwn.cdc.gov/nchs/nhanes/Default.aspx, accessed on 31 August 2021).

### 2.2. Estimation of Dietary Intakes

The dietary intake assessment was based on the data collected in the dietary recall component of NHANES known as What We Eat in America (WWEIA, details in https://www.cdc.gov/nchs/nhanes/wweia.htm, accessed on 31 August 2021). Due to concerns about data quality, only the data from the first 24-hour dietary recall were used in this study to estimate the dietary and protein intake.

Proteins were estimated based on cycle-specific versions of the US Department of Agriculture Food and Nutrition Database for Dietary Studies. We defined animal protein as protein from animal products, including dairy products, red meat, poultry, fish/shellfish, and eggs. In contrast, plant protein was defined as protein from plant products, including beans, nuts, grains, vegetables, and fruit. Some foods contain both animal and plant ingredients, which are separately recorded as mixed food. Energy from mixed foods is included in the calculation of total energy. However, the protein from mixed foods is only included in the calculation of total protein and is not included in the calculation of animal and plant protein.

Estimates of total protein intake, animal protein intake, and plant protein intake were first generated as grams per day. Using the residual method to minimize measurement error in dietary estimates, the absolute intakes of proteins in grams per day were adjusted for total energy intake to the median level of the study population. After minimizing the measurement error, we calculated the ratio of animal protein intake to plant protein intake, which is recorded as the AP ratio. Then all kinds of protein intake and AP ratio were divided into three levels by using tertiles.

### 2.3. Assessment of Glucose Homeostasis Indicators

The outcomes were the indicators of glucose homeostasis, including fasting glucose (GLU) and insulin (INS). Insulin resistance was calculated using the homeostasis model assessment of insulin resistance (HOMA-IR). The following formula was used:$$HOMA-IR=INS\;\left(mIU\times L^{-1}\right)\times GLU\;\left(mmol\times L^{-1}\right)÷22.5$$

### 2.4. Adjustment of Covariates

Since body mass index (BMI) was considered to be an important indicator affecting insulin resistance, pre-pregnancy BMI was included in the statistical analysis as a covariate. Though the standing height, weight, and calculated BMI were provided in the Examination part of NHANES, it only reflected the BMI during pregnancy. Therefore, we recalculated the pre-pregnancy BMI based on the self-reported weight one year ago in the Questionnaire Data and divided it into three levels according to 18.5 and 24.0 as the cutoff values.

Due to the different physiological changes in glucose metabolism in different gestation periods, the gestation period was considered to be another important covariate. It was also described with three levels, namely the first, second, and third trimesters.

Other demographic characteristics, including parity, age, race, education, and family income, were considered potential covariates. Only when the distribution of outcome indicators among subjects with different demographic characteristics was statistically different, the corresponding demographic characteristics would be included in the regression model.

### 2.5. Animal Experiment

Thirty-six specific-pathogen-free twelve-week-old male Sprague Dawley (SD) rats were provided by the Department of Laboratory Animal Science of Peking University (Beijing, China, SCXK2016-0010). The SD rats were randomly assigned to three groups of twelve rats each, including a 100% animal protein group, a 50% animal protein group, and a 100% plant protein group. Animals were housed with an automatically controlled temperature (24 ± 0.5 °C) and humidity (50 ± 10%) and a 12-hour light–dark cycle with free access to food and water.

The total experimental duration was about three weeks. After three days of adaptive feeding, the female rats were mated with males (females:males = 2:1) at 6:00 pm. The day when spermatozoa were identified was considered day 0.5 of pregnancy, and the rats identified as pregnant were transferred to individual cages. Then three groups were fed different diets.

This study used different kinds of diets based on the AIN-93G diet. Three different diets were prepared: (1) 100% animal protein diet (20% weight as pure animal protein from milk protein concentrate), (2) 50% animal protein diet (10% weight as pure animal protein from milk protein concentrate, and 10% weight as pure plant protein from soy protein isolate), (3) 100% plant protein diet (20% weight as pure plant protein from soy protein isolate). All these diets contained the same caloric density and macronutrient compositions. The content of nutrients in the feed is shown in Table 1.

After overnight fasting on the evening of day 18.5 of pregnancy, these pregnant rats were anesthetized with pentobarbital sodium (150 mg/kg) in the morning of day 19.5 of pregnancy. Meanwhile, the blood was collected from the abdominal aorta, and plasma was obtained and stored in a −80℃ ultralow-temperature refrigerator.

The protocol of the animal experiment was reviewed and approved by the Animal Ethical Committee at Peking University (Protocol #: LA2020440). All animals were handled in accordance with the guidelines of the Peking University Animal Research Committee (www.lab.pku.edu.cn, accessed on 31 August 2021) and the National Institutes of Health (NIH Publication No. 85-23 revised 1985).

### 2.6. Laboratory Testing of Animal Experiment

The levels of GLU and INS in the plasma of pregnant rats were detected by the enzyme microplate method, and HOMA-IR was calculated by GLU × INS÷22.5.

In metabolomics testing, 100 μL of plasma was accurately drawn after thawing at 4 °C, and 300 μL of cold acetonitrile was added. Then the suspension was ultrasonically extracted for 30 min in an ice bath and centrifuged at 12,000 rpm for 10 min at 4 °C, followed by 100 μL being removed and concentrated to dryness by vacuum centrifugation at 37 °C. The residue was dissolved with 100 μL acetonitrile and centrifuged at 12,000 rpm and 4 degrees Celsius for 10 min. Finally, 10 μL of the supernatant was injected and analyzed by ultrahigh-performance liquid chromatography–mass spectrometry (UPLC-MS).

### 2.7. Statistical Methods

Data analysis was performed using R4.1.2. The Kruskal–Wallis rank-sum test was used to evaluate whether the distribution of outcome indicators was different among subjects with different demographic characteristics. After adjusting for total energy intake using the residual method, multiple linear regression was used to observe the association of total protein intake, animal protein intake, plant protein intake, and AP ratio with the outcomes, adjusting the potential confounding factors, including pre-pregnancy BMI, gestation period, and demographic characteristics. Considering the different physiological changes in glucose metabolism in different gestation periods, a stratified analysis was further carried out according to the gestation period.

In the animal study, ANOVA was used to compare whether the levels of GLU, INS, and HOMA-IR were different between the three experimental groups, and trend variance analysis was performed to test for trends. Sources of differences between groups were analyzed by using the HSD post hoc test.

The original metabolomic data were converted into ABF format by Analysis Base File Converter software and imported into MS-DIAL 4.60 for preprocessing to obtain the original data matrix. The MassBank, Respect, and GNPS databases were searched, and the extracted peak information was compared with the databases to identify the metabolite species contained in the sample.

Principal component analysis (PCA) was used to observe the overall metabolic profile of each sample and the natural distinction between samples. Then partial least-squares discrimination analysis (PLS-DA) was used to distinguish the overall difference in metabolic profile between groups, calculate the variable importance for the projection (VIP) of each metabolite, and judge the overfitting of the PLS-DA model by permutation test. At the same time, the fold change (FC) of metabolites between different groups was calculated, and the *t*-test was used to determine whether there was a statistical difference in metabolite levels between groups. Based on the results of the VIP value, FC, and *t*-test, the differential metabolites between the experimental groups were screened out. The screening criteria in this study were as follows: VIP > 1, FC > 1.5 or <2/3, and *p*-value < 0.05 in the *t*-test. After screening out the differential metabolites between groups, KEGG metabolic pathway analysis was used to determine the enrichment of differential metabolites in known metabolic pathways, and the BH method was used for multiple correction tests in KEGG analysis.

## 3. Results

### 3.1. Characteristics of the Study Population

Table 2 shows the characteristics of the study population. There were statistically significant differences in the distribution of all outcome indicators among the subjects at different gestational periods. In addition, there were statistical differences in the distribution of GLU and INS among subjects of different ages and HOMA-IR among different races.

### 3.2. Results of the Population Study

The multiple linear regression results of total protein intake and GLU, INS, or HOMA-IR are shown in Table 3. There was no statistical association between total protein intake and GLU, INS, or HOMA-IR. The multiple linear regression results of animal protein, plant protein intake, and AP ratio are shown in Table 4. There was a negative correlation between plant protein intake and INS or HOMA-IR after adjusting for the pre-pregnancy BMI, gestational period, and demographic characteristics, including age and race.

After stratification by gestational period, animal protein intake was not associated with GLU, INS, or HOMA-IR. In the third trimester, plant protein intake was negatively correlated with HOMA-IR, and there may still be some correlation between plant protein and INS, but it is not significant enough (*p* < 0.10). Interestingly, the AP ratio showed a positive correlation with HOMA-IR in the third trimester, while the correlation with INS was not significant (*p* < 0.10). The results are shown in Table 5.

### 3.3. Results of the Animal Experiment

As shown in Table 6, there were significant differences in the levels of INS and HOMA-IR among the three groups, and the trend test results showed that the levels of INS and HOMA-IR decreased with the increase in plant protein in the diet. Figure 2 shows that the differences were mainly from the 100% animal protein and 100% plant groups. Therefore, a subsequent metabolomic analysis was carried out between these two groups.

We first examined the overall metabolic profiles of the plasma of the two groups of pregnant rats using PCA, and the result is shown in Figure 3. In the space of the first three principal components, the metabolic profiles of the two groups of pregnant rats were clearly differentiated, indicating that the plasma of the two groups of pregnant rats could be naturally distinguished.

We further analyzed the differential metabolites between the plasma of the two groups of pregnant rats. Only the metabolites that satisfied all three conditions, namely VIP > 1, FC > 1.5 or <2/3, and *p*-value < 0.05 in the *t*-test, at the same time were considered differential metabolites. We also excluded exogenous drug and phytochemical components, and the results of endogenous differential metabolites for the top 25 VIP values are shown in Table 7. These endogenous differential metabolites mainly include a variety of amino acids, vitamins, and their derivatives. In addition to the metabolites shown in the table, endogenous differential metabolites also contain a variety of other substances, such as taurine, betaine, acetylcarnitine, and glycocholic acid.

The results of the KEGG metabolic pathway analysis are presented in Table 8. After the intervention of different protein sources, multiple differential metabolic pathways were found between the two groups. These differential metabolic pathways include a variety of amino acid metabolism pathways, vitamin metabolism pathways, some signaling pathways that affect hormone secretion, such as the prolactin signaling pathway, and some pathways considered key pathways that affect energy and substance metabolism, such as the FoxO and mTOR signaling pathways.

## 4. Discussion

In this study, by combining a population study and an animal experiment, we found that the intake of different source proteins during pregnancy may affect maternal insulin resistance. In the population study, plant protein intake was negatively correlated with INS and HOMA-IR. The animal experiment obtained consistent results with the population study and showed that the levels of INS and HOMA-IR decreased with the increase in the proportion of plant protein intake in pregnant rats in late pregnancy, revealing that the sources of dietary protein affect maternal insulin resistance. Because of the known progressive increase in insulin resistance and the increased dietary protein requirements throughout pregnancy, the relationship between dietary protein and insulin resistance during pregnancy is important. Therefore, clarifying the relationships between dietary protein intake and maternal insulin resistance will contribute to understanding the optimal protein source and requirements for adequate maternal and fetal needs without compromising the health of the pregnant female or her offspring.

We found that there was a relationship between plant protein intake and insulin resistance in pregnant females, which may reveal the possibility of the association between plant protein intake and glucose metabolism. Some studies used a model or an intervention to substitute animal protein with plant protein and reported mitigation of the development of insulin resistance [37,38], suggesting the beneficial effect of plant protein on pregnant females. Different from other studies of an increased risk of gestational diabetes mellitus in the pregnant population and increased insulin resistance in the non-pregnant population [24,39,40], this current study did not find any relationship between total protein and animal protein intake and insulin resistance.

To our knowledge, the only previous study considering the effects of the amount or source of dietary protein intake on maternal insulin resistance was performed by Allman and colleagues [41]. Their research was based on 173 pregnant females in late pregnancy from Arkansas (United States) and found that there was a negative relationship between total protein and plant protein intake and insulin resistance in females in late pregnancy, yet no relationship existed between animal protein intake and insulin resistance, partially validating our findings [41]. Compared with their study, our study is not limited to pregnant females in late pregnancy and the study subjects selected are more representative; it is not surprising that fully consistent results were not obtained considering the complex background of the population.

In population studies, it is often difficult to completely exclude the confounding effects of different ethnic backgrounds, lifestyles, and underlying disease states on insulin resistance, and the estimation of dietary intake may also be biased. Although the influence of total dietary energy intake and demographic factors has been excluded as much as possible, the current observational study can only find etiological associations, and only randomized controlled trials can give rigorous etiological inferences. Therefore, animal experiments were designed to verify the results found in the population study.

In the animal experiment, we found results that were consistent with the population study; maternal INS and HOMA-IR decreased with the increase in dietary plant protein intake and proportion. To our knowledge, studies in animals previously were only interested in the amount of dietary protein [42,43], and no animal studies have investigated the effect of different dietary protein sources on maternal insulin resistance. Our animal experiment validated clues found in the population study and further revealed that the source of dietary protein during pregnancy affected the maternal plasma metabolome, contributing to clarifying the mechanism of different sources of dietary protein affecting insulin resistance during pregnancy.

Under a certain amount of total protein intake in our study, the levels of various metabolites in the plasma of pregnant rats between the 100% plant protein group and the 100% animal protein group were significantly changed, including amino acids and their derivatives, vitamins and their derivatives, fatty acid derivatives, carboxylic acids, and other active metabolites. The reduction in BCAAs in the peripheral circulation is thought to be associated with a reduction in insulin resistance [44,45]. In this study, we found that compared with the 100% animal protein group, leucine, which was considered the most representative amino acid among BCAAs, decreased significantly in the plasma of the 100% plant protein group, which may be one of the mechanisms by which the intake of plant protein reduces the maternal insulin resistance. Studies based on non-pregnant people have found that histidine intake restriction may be associated with a decrease in insulin resistance [46], and a significant reduction in histidine levels was also found in the 100% plant protein group. In addition to the above two amino acids, a variety of other active metabolites may also affect maternal insulin resistance by affecting metabolic pathways. Among them, the mTOR signaling pathway is an important pathway for the body’s regulation of anabolic and catabolic processes in response to environmental signals [47]. In this study, we found that in addition to leucine, the levels of various signaling molecules such as adenosine diphosphate (AMP), arginine, glutamine, and asparagine were significantly changed in the mTOR signaling pathway. AMP affected the mTOR signaling pathway by affecting the activity of AMP-activated protein kinase (AMPK) [48]. Signaling from arginine was transduced to mTORC1 through Rag GTPase [49,50], and glutamine or asparagine could affect the mTOR signaling pathway independently of Rag GTPase [51,52]. Changes in the levels of the above active substances may have an impact on the maternal mTOR signaling pathway, thereby affecting maternal insulin sensitivity. The FoxO signaling pathway is another important pathway that may affect maternal insulin resistance, which can affect insulin signal transduction and insulin sensitivity of the body, and mediate the effects of insulin or insulin-like growth factors on key functions such as cell metabolism, growth, differentiation, oxidative stress, aging, and autophagy [53]. In this study, the differential metabolite changes related to the FoxO signaling pathway also included glutamate and AMP as signaling molecules in the upstream pathway. In addition, tyrosine and its metabolites can affect the prolactin signaling pathway. Prolactin is a protein hormone secreted by the pituitary gland with complex biological roles [54]. Changes in the prolactin signaling pathway may affect maternal hormone secretion and balance, thereby affecting insulin resistance during pregnancy.

In epidemiological studies, it is always difficult to discuss the health benefits of plant-based foods separately from plant proteins, and there is also much debate as to whether the health benefits of plant-based foods stem from their protein content [55]. In this study, by combining the population study and the animal experiment, we avoid the shortcomings mentioned before well. In addition, the establishment of animal models also excluded the confounding of the remaining nutrients in the diet and supplemented the evidence on long-term dietary intake. Meanwhile, the effect of total energy was well balanced in our study, making the evidence more credible.

There are several limitations in this study. First, this study only studied the US population; as we all know, the dietary habits of distinct populations are very different, and the evidence found in this study should be validated in studies from other regions. Second, as the limitation of the data source, we were unable to obtain long-term dietary information for the population; though animal experiments partially make up for this deficiency, long-term observations of humans in the future would be better. Finally, randomized controlled dietary interventions can be conducted in populations rather than animals to provide more convincing evidence.

## 5. Conclusions

We found a negative association between dietary plant protein intake and maternal insulin resistance during pregnancy. Different protein intake could induce changes in some amino acids and other active substances in plasma, as well as related metabolic pathways such as the mTOR signaling pathway and FoxO signaling pathway, which may affect maternal insulin resistance.

## Figures and Tables

**Figure 1 nutrients-14-05039-f001:**
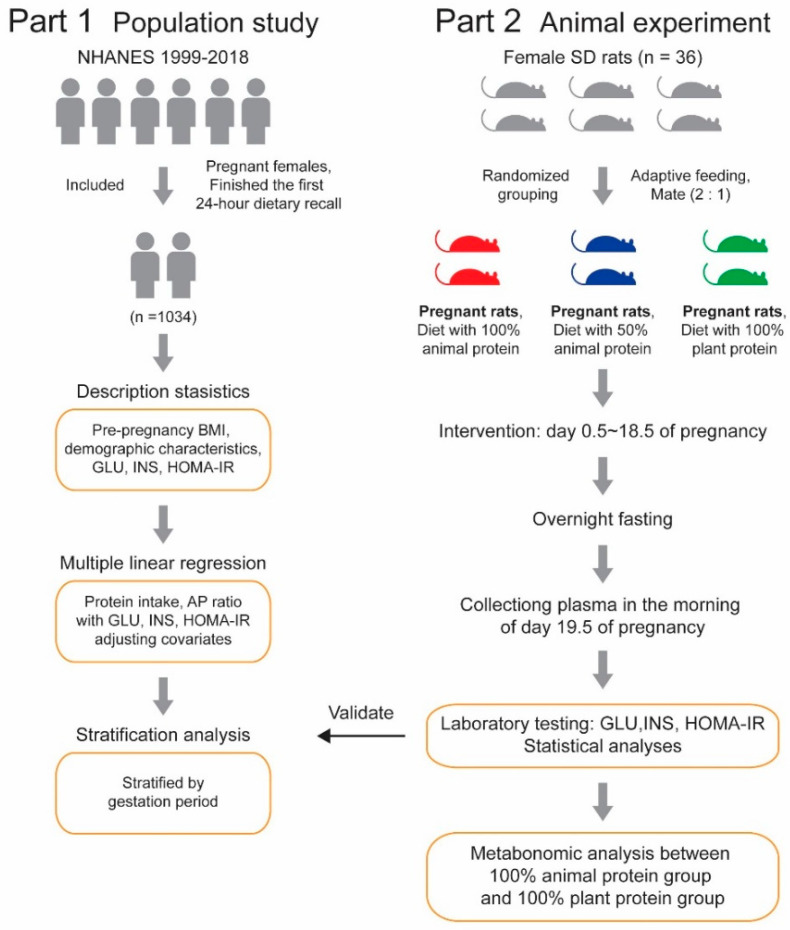
Design of this study.

**Figure 2 nutrients-14-05039-f002:**
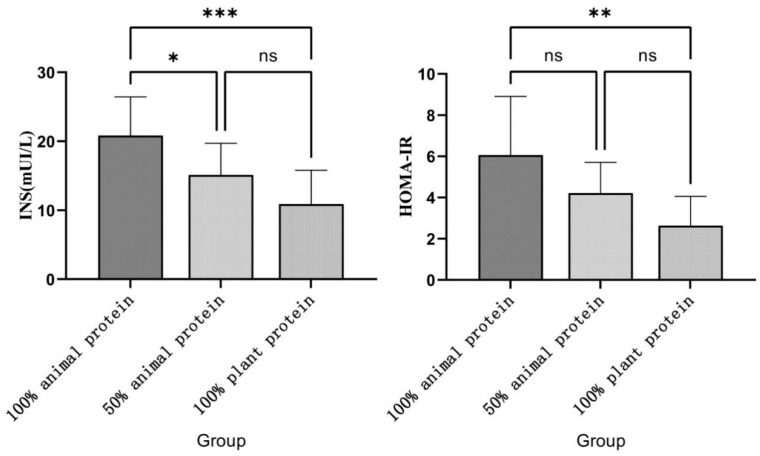
INS and HOMA-IR levels in the three groups. Figure legend: The INS and HOMA-IR levels in the 100% animal protein, 50% animal protein, and 100% plant protein group. Compared to the 100% animal protein group, HOMA-IR and insulin levels were significantly decreased in the 100% animal protein group. At the same time, the insulin level of the 50% animal protein group was also significantly lower than that of the 100% animal protein group. * adjusted *p* < 0.05; **, adjusted *p* < 0.01; *** adjusted *p* < 0.001.

**Figure 3 nutrients-14-05039-f003:**
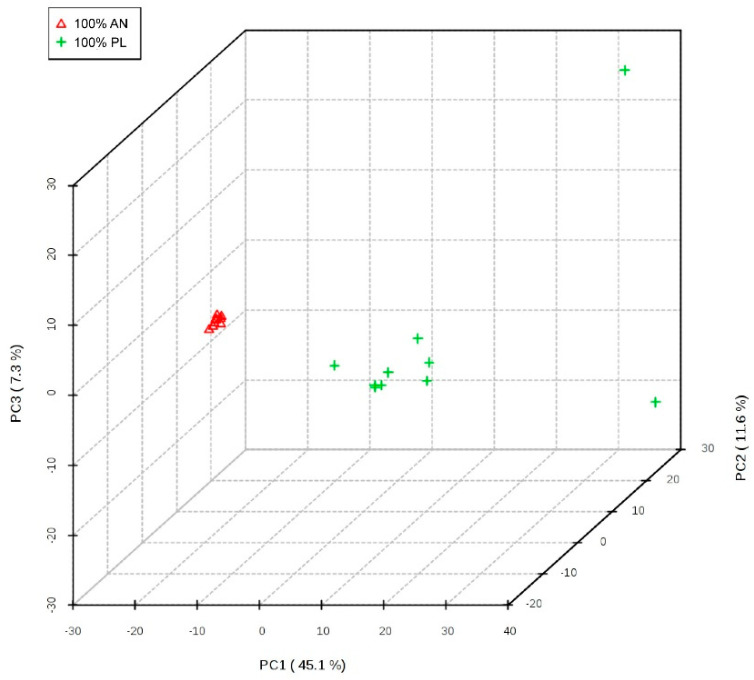
The overall metabolic profiles of the plasma of the two groups. Figure legend: The overall metabolic profiles of the plasma of the two groups of pregnant rats were examined using PCA. The metabolic profiles of the two groups of pregnant rats could be naturally distinguished. 100% AN: 100% animal protein group, 100% PL: 100% plant protein group, PC: principal component.

**Table 1 nutrients-14-05039-t001:** Composition of the experimental diets.

Component (g/kg) ^a^	100% Animal Protein	50% Animal Protein	100% Plant Protein
Total energy (kcal/kg) ^a^	3832.48	3832.48	3832.48
Protein	200.00	200.00	200.00
Fat	70.00	70.00	70.00
Carbohydrates	600.62	600.62	600.62
Dietary fiber	50.00	50.00	50.00
Milk protein concentrate	241.08	120.54	-
Soy protein isolate	-	110.07	220.14
L-cystine	3.00	3.00	3.00
Corn starch	360.35	361.67	362.98
Maltodextrin	132.00	132.00	132.00
Sucrose	100.00	100.00	100.00
Cellulose	50.00	49.86	49.71
Oil	66.07	68.01	69.96
Mineral mix	35.00	35.00	35.00
Vitamin mix	10.00	10.00	10.00
Choline bitartrate	2.50	2.50	2.50
BHT ^b^	0.01	0.01	0.01
Distilled water	0.00	7.35	14.70

^a^. All units are g/kg except for the total energy, which is kcal/kg. ^b^. BHT: dibutylhydroxytoluene, a food antioxidant.

**Table 2 nutrients-14-05039-t002:** Characteristics of study population ^a^.

Characteristics	n (%)	GLU (mmol/L)	INS (pmol/L)	HOMA-IR
All subjects	1034 (100)	4.61 (4.31, 4.96)	57.0 (38.6, 94.4)	2.01 (1.31, 3.29)
pre-pregnancy BMI				
<18.5	22 (2.1)	4.28 (3.98, 4.59) ^c^	29.7 (19.6, 44.8) ^c^	0.88 (0.60, 1.55) ^c^
18.5–23.9	193 (18.7)	4.56 (4.29, 4.84) ^c^	46.9 (34.1, 74.2) ^c^	1.60 (1.14, 2.41) ^c^
≥24.0	252 (24.4)	4.70 (4.37, 5.05) ^c^	69.4 (47.2, 107.6) ^c^	2.32 (1.65, 3.84) ^c^
Gestation period				
First	150 (14.5)	4.76 (4.55, 5.05) ^c^	48.4 (37.0, 89.0) ^c^	1.77 (1.29, 3.18) ^c^
Second	305 (29.5)	4.49 (4.22, 4.84) ^c^	53.5 (37.2, 84.4) ^c^	1.84 (1.19, 2.65) ^c^
Third	277 (26.8)	4.47 (4.22, 4.79) ^c^	73.6 (53.6, 108.9) ^c^	2.40 (1.81, 3.76) ^c^
Parity				
Primiparous	42 (4.1)	4.71 (4.54, 5.02)	49.2 (39.1, 72.1)	1.76 (1.26, 2.56)
Non-primiparous	675 (65.3)	4.61 (4.31, 4.95)	55.2 (38.8, 87.0)	1.91 (1.30, 2.91)
Age				
20–24	320 (30.9)	4.51 (4.22, 4.88) ^c^	62.1 (40.1, 100.7) ^c^	2.15 (1.37, 3.43)
25–29	313 (30.3)	4.59 (4.34, 4.86) ^c^	62.0 (39.8, 89.9) ^c^	2.03 (1.30, 3.28)
30–34	257 (24.9)	4.65 (4.31, 5.11) ^c^	61.0 (38.3, 98.2) ^c^	2.18 (1.35, 3.68)
35–39	122 (11.8)	4.78 (4.53, 5.11) ^c^	48.1 (33.2, 62.8) ^c^	1.80 (1.14, 2.33)
40–44	22 (2.1)	4.83 (4.59, 5.11) ^c^	37.4 (28.7, 50.5) ^c^	1.28 (1.02, 1.83)
Race				
Mexican	250 (24.4)	4.63 (4.27, 5.03)	66.6 (45.5, 102.5) ^c^	2.22 (1.56, 3.63) ^c^
Other Hispanic	73 (7.1)	4.69 (4.38, 5.00)	70.4 (48.8, 100.7) ^c^	2.31 (1.81, 3.28) ^c^
Non-Hispanic White	445 (43.5)	4.61 (4.33, 4.88)	50.3 (33.3, 79.1) ^c^	1.77 (1.12, 2.69) ^c^
Non-Hispanic Black	165 (16.1)	4.57 (4.23, 4.98)	62.0 (40.1, 97.3) ^c^	2.10 (1.32, 3.47) ^c^
Other race	101 (9.9)	4.66 (4.38, 5.11)	58.8 (39.2, 88.1) ^c^	2.09 (1.25, 3.28) ^c^
Education				
Less than 9th grade	76 (7.4)	4.61 (4.23, 4.94)	53.5 (40.6, 90.8)	1.90 (1.32, 3.28)
9–11th grade	176 (17.0)	4.57 (4.28, 4.86)	66.7 (46.9, 104.1)	2.28 (1.69, 3.53)
High school graduate	214 (20.7)	4.62 (4.31, 5.00)	62.1 (40.1, 104.5)	2.19 (1.37, 3.76)
AA degree	294 (28.4)	4.63 (4.33, 4.91)	55.6 (37.9, 91.3)	1.96 (1.27, 3.13)
College graduate or above	273 (26.4)	4.61 (4.33, 5.00)	53.2 (33.4, 80.6)	1.83 (1.10, 2.76)
Family income ^b^				
<1.00	244 (23.6)	4.61 (4.31, 4.94)	55.6 (39.2, 90.6)	1.98 (1.32, 3.15)
1.00–2.99	213 (20.6)	4.61 (4.28, 5.00)	56.0 (39.1, 94.5)	1.90 (1.31, 3.32)
3.00–4.99	129 (12.5)	4.59 (4.35, 4.93)	62.2 (40.9, 91.5)	2.09 (1.31, 3.37)
≥5.00	177 (17.1)	4.64 (4.35, 5.05)	49.2 (33.3, 81.6)	1.75 (1.10, 2.90)

^a^. Described by median and upper and lower quartiles (P_25_, P_75_); ^b^. Family income is represented by the ratio of annual family income to the US poverty line, and if <1.00, the research subject is considered to be in a poor group. ^c^. There was a statistically significant difference in the distribution of the corresponding outcomes within the group, *p* < 0.05.

**Table 3 nutrients-14-05039-t003:** The multiple linear regression results of total protein intake and GLU, INS, or HOMA-IR.

Variables			GLU	INS	HOMA-IR
Total protein intake	Model 1 ^a^	β (95%CI)	0.01 (−0.06, 0.08)	8.74 (−4.58, 22.07)	0.46 (−0.19, 1.10)
		*p*-value	0.724	0.198	0.168
	Model 2 ^b^	β (95%CI)	−0.01 (−0.08, 0.06)	5.47 (−11.85, 22.78)	0.28 (−0.56, 1.12)
		*p*-value	0.704	0.535	0.513

^a^. Model 1: Unadjusted for covariates. ^b^. Model 2: Adjusted for pre-pregnancy BMI, gestation period, and demographic characteristics, including age and race.

**Table 4 nutrients-14-05039-t004:** The multiple linear regression results of animal protein intake, plant protein intake, AP ratio, and GLU, INS, or HOMA-IR.

Variables			GLU (mmol/L)	INS (mIU/L)	HOMA-IR
animal protein intake	Model 1 ^a^	β (95%CI)	−0.09 (−0.16, −0.02)	7.31 (−6.02, 20.65)	0.31 (−0.34, 0.96)
		*p*-value	0.015	0.282	0.351
	Model 2 ^b^	β (95%CI)	−0.06 (−0.13, 0.01)	3.49 (−14.12, 21.10)	0.20 (−0.67, 1.05)
		*p*-value	0.103	0.696	0.667
plant protein intake	Model 1 ^a^	β (95%CI)	−0.03 (−0.10, 0.04)	−18.73 (−31.97, −5.49)	−0.96 (−1.60, −0.32)
		*p*-value	0.398	0.006	0.004
	Model 2 ^b^	β (95%CI)	0.00 (−0.06, 0.07)	−20.60 (−37.91, −3.30)	−1.04 (−1.89, −0.20)
		*p*-value	0.892	0.020	0.015
AP ratio	Model 1 ^a^	β (95%CI)	−0.03 (−0.10, 0.04)	12.39 (−0.91, 25.69)	0.63 (−0.02, 1.27)
		*p*-value	0.360	0.068	0.058
	Model 2 ^b^	β (95%CI)	−0.05 (−0.12, 0.02)	13.91 (−3.66, 31.48)	0.73 (−0.13, 1.59)
		*p*-value	0.175	0.120	0.096

^a^. Model 1: Unadjusted for covariates. ^b^. Model 2: Adjusted for pre-pregnancy BMI, gestation period, and demographic characteristics, including age and race.

**Table 5 nutrients-14-05039-t005:** The multiple linear regression results of plant protein intake, AP ratio, and INS or HOMA-IR after stratification by gestation period ^a^.

Variables	Gestation Period		INS (mIU/L)	HOMA-IR
Plant protein intake	First trimester	β (95%CI)	−26.23 (−60.73, 8.27)	−1.25 (−2.88, 0.39)
		*p*-value	0.133	0.133
	Second trimester	β (95%CI)	−4.03 (−19.77, 11.71)	−0.19 (−0.81, 0.43)
		*p*-value	0.614	0.541
	Third trimester	β (95%CI)	−38.47 (−76.99, 0.04)	−2.10 (−4.04, −0.15)
		*p*-value	0.050	0.035
AP ratio	First trimester	β (95%CI)	18.69 (−15.97, 53.34)	0.93 (−0.70, −2.57)
		*p*-value	0.285	0.258
	Second trimester	β (95%CI)	3.96 (−11.53, −19.45)	0.12 (−0.50, 0.73)
		*p*-value	0.614	0.707
	Third trimester	β (95%CI)	36.76 (−2.06, 75.57)	1.96 (0.01, 3.91)
		*p*-value	0.063	0.048

^a^. Adjusted for pre-pregnancy BMI and demographic characteristics, including age and race.

**Table 6 nutrients-14-05039-t006:** GLU, INS, and HOMA-IR levels in rats administered different proportions of protein.

Group	100% Animal Protein Mean ± sd	50% Animal Protein Mean ± sd	100% Plant Protein Mean ± sd	*p* for ANOVA	*p* for Trend
GLU (mmol/L)	6.40 ± 2.15	6.25 ± 1.11	5.61 ± 1.74	0.557	0.314
INS (mIU/L)	20.83 ± 5.62	15.09 ± 4.60	10.88 ± 4.91	0.001	0.000
HOMA-IR	6.06 ± 2.85	4.22 ± 1.49	2.63 ± 1.42	0.007	0.002

**Table 7 nutrients-14-05039-t007:** Endogenous differential metabolites between the 100% plant protein group and 100% animal protein group.

Metabolites	log_2_FC ^a^	*p*-Value	VIP
L-Tyrosine	−9.19	<0.000	1.55
L-Histidine	−5.37	<0.000	1.55
L-asparagine	−4.73	<0.000	1.55
5-Aminolevulinic acid	−7.41	<0.000	1.55
Phenylalanine	−6.43	<0.000	1.55
Glutamine	−3.41	<0.000	1.55
Threonine	−2.71	<0.000	1.54
Succinic acid	−2.04	<0.000	1.50
N-Acetylaspartate	−6.99	<0.000	1.50
2-hydroxy-6-methylisonicotinic acid	3.97	<0.000	1.50
N, N-Dimethylarginine	6.42	<0.000	1.49
Arginine	6.14	<0.000	1.49
Gly	3.85	<0.000	1.46
1-Methylhistamine	−3.80	<0.000	1.45
Pantothenate	−7.00	<0.000	1.45
L-Leucine	−5.49	<0.000	1.44
L-Glutamic acid	−3.63	<0.000	1.43
L-Citrulline	3.87	<0.000	1.43
D-Glucosamine-6-phosphate	6.71	<0.000	1.42
N-Methyl-L-proline	5.80	<0.000	1.37
Glutamic acid	1.97	<0.000	1.35
Thiamine	3.98	<0.000	1.33
Lysine	6.59	<0.000	1.33
Riboflavin	5.29	<0.000	1.30
N8-Acetylspermidine	3.27	<0.000	1.30

^a^. The 100% plant protein group vs. the 100% animal protein group.

**Table 8 nutrients-14-05039-t008:** Differential metabolic pathways between 100% plant protein group and animal protein group.

Pathway Name	KEGG Pathway ID	Adjusted *p*-Value
Biosynthesis of amino acids	rno01230	<0.000
Alanine, aspartate, and glutamate metabolism	rno00250	<0.000
GABAergic synapse	rno04727	<0.000
Arginine biosynthesis	rno00220	0.000
FoxO signaling pathway	rno04068	0.000
Glutamatergic synapse	rno04724	0.002
Glyoxylate and dicarboxylate metabolism	rno00630	0.003
Prolactin signaling pathway	rno04917	0.004
Histidine metabolism	rno00340	0.005
D-Glutamine and D-glutamate metabolism	rno00471	0.006
Glycine, serine, and threonine metabolism	rno00260	0.006
mTOR signaling pathway	rno04150	0.007
Tyrosine metabolism	rno00350	0.008
Phenylalanine metabolism	rno00360	0.012
Pyrimidine metabolism	rno00240	0.014
Purine metabolism	rno00230	0.015
Bile secretion	rno04976	0.016
Taurine and hypotaurine metabolism	rno00430	0.016
cAMP signaling pathway	rno04024	0.022
Cholesterol metabolism	rno04979	0.028
Phospholipase D signaling pathway	rno04072	0.032
Thiamine metabolism	rno00730	0.033

## Data Availability

The data used for the population study in this study are publicly available from https://www.cdc.gov/nchs/nhanes/index.htm (accessed on 31 August 2021). For the data in the animal experiment, please contact the authors directly.

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
