# Peer review of "Dietary Plant Protein Intake Can Reduce Maternal Insulin Resistance during Pregnancy"

_nutrients, 2022, doi:10.3390/nu14235039_

Round 1

Reviewer 1 Report

This study in pregnant women aims to compare the effects of protein intake from animal vs. plant sources on insulin values and insulin resistance by combining population and animal studies. 

The paper addresses an interesting and topical issue. There are strong discrepancies among nutrition professionals on what is the correct diet in pregnancy to reduce the risk of insulin resistance or diabetes mellitus.

These studies were conducted in a formally correct manner. However, there are some issues that need to be resolved before possible publication. 

1. There is plagiarism between the methods, please check the attached file

2. It is not very clear why two studies were done. In particular, the work on rats appears to be taken out of context. The mice are not pregnant so why should the data on the effects of the 3 different diets be complementary to the data in NHANES which are on pregnant women? 

3. The metabolomic analysis appears to be the most innovative part of the paper. It could help explain why plant-based diets have advantages over animal-based diets. However, the data are not well explained, they need to be further investigated.

4. There is little nutrition in this paper. What are the differences in foods between animal protein and plant protein? References should be added to studies that have evaluated the different effects of different foods of animal or plant origin on the risk of diabetes.

Author Response

We are very grateful to the reviewer for reviewing our article. We have summarized the comments and our responses in the following document.

Reviewer 2 Report

Article is very good and interesting and the outcome and results are important to advise pregnant mothers but probably also to advise others f.i. elderly. 

I found a few typo's. Line 40 and 41: it is a bit confusing that you relate insulin resistance to intra-uterine growth retardation and large for gestational age newborns? 

Table 2: Primitive must be primiparous. Also non-primitive must be non-primiparous. 

Line 361 "aging" is double mentioned. 

Line 376 serval must be several.  

Author Response

(The authors gave the same response as above.)

Round 2

Reviewer 1 Report

The authors fixed the apper following my instructions

There remains a strong doubt (also ethical) about the usefulness of using the animal model in this study.